# Immune Response to Chikungunya Virus: Sex as a Biological Variable and Implications for Natural Delivery via the Mosquito

**DOI:** 10.3390/v15091869

**Published:** 2023-09-03

**Authors:** Meagan Taylor, Jonathan O. Rayner

**Affiliations:** Department of Microbiology & Immunology, Whiddon College of Medicine, University of South Alabama, Mobile, AL 36688, USA; mmt1322@jagmail.southalabama.edu

**Keywords:** chikungunya virus, vector-borne diseases, *Aedes*, sex differences, TLR7/8

## Abstract

Chikungunya virus (CHIKV) is a mosquito-borne virus with significant public health implications around the world. Climate change, as well as rapid urbanization, threatens to expand the population range of *Aedes* vector mosquitoes globally, increasing CHIKV cases worldwide in return. Epidemiological data suggests a sex-dependent response to CHIKV infection. In this review, we draw attention to the importance of studying sex as a biological variable by introducing epidemiological studies from previous CHIKV outbreaks. While the female sex appears to be a risk factor for chronic CHIKV disease, the male sex has recently been suggested as a risk factor for CHIKV-associated death; however, the underlying mechanisms for this phenotype are unknown. Additionally, we emphasize the importance of including mosquito salivary components when studying the immune response to CHIKV. As with other vector-transmitted pathogens, CHIKV has evolved to use these salivary components to replicate more extensively in mammalian hosts; however, the response to natural transmission of CHIKV has not been fully elucidated.

## 1. Introduction

Chikungunya virus (CHIKV) is a mosquito-borne arbovirus that spreads to mammalian hosts via the bite of an infected mosquito, predominantly *Aedes aegypti* and *Aedes albopictus* in urban cycles [1,2,3]. Like several *Aedes*-borne diseases (i.e., dengue virus and Zika virus), CHIKV is an emerging public health threat. Currently, there are no approved vaccines or therapeutics to prevent or treat CHIKV (although a VLP-based vaccine is in phase 3 clinical trials at the time of this article). As climate change continues to increase temperatures globally, the regions in which *Ae. aegypti* and *Ae. albopictus* can thrive are expanding [4]. Additionally, international trade and rapid urbanization have spread mosquito eggs to previously unoccupied areas, increasing their population numbers in these new regions [3,5]. With the threat of climate change, the expansion of these vector habitats is imminent. As a result, CHIKV poses an increasing risk of human infection worldwide.

CHIKV was first isolated in the early 1950s in East Africa and continues to cause outbreaks in Africa and Southeast Asia [2,3]. “Chikungunya” translates to “that which bends” in the African Makonde dialect, in reference to the bent posture exhibited by patients with CHIKV arthralgia. CHIKV infections are associated with a wide range of symptoms, from fever and rash to encephalitis or chronic polyarthralgia. CHIKV has very few asymptomatic cases. Moreover, as few as 3% of cases may be asymptomatic when infected with CHIKV, while as many as 80% of cases report persistent and debilitating arthralgia for months or even years after the initial infection [2,6]. Remarkably, data from recent CHIKV outbreaks suggest an increase in the severity of the infection symptoms. While case fatality rates are estimated to be 1 death per 1000 cases, the first documented deaths from CHIKV were reported during the 2005–2006 outbreak in La Reúnion [2,7,8]. Though the first documented deaths of CHIKV appear to suggest worsening symptomology associated with its spread to previously unexposed areas, reporting biases also play a role. For example, CHIKV outbreaks tend to receive more attention than endemic cases of CHIKV. As a result, the areas from which CHIKV originated, such as the African continent, have not been as thoroughly monitored. This presents a large gap in knowledge regarding vector-borne disease transmission and symptomology and should be addressed expeditiously.

Intriguingly, recent epidemiological studies also suggest that the female sex is a risk factor for severe or chronic CHIKV symptomology despite evidence that females generally mount a more robust immune response than males. In contrast, some studies indicate that male sex is a risk factor for CHIKV-associated deaths [9]. Female patients in a South Indian outbreak reported significantly more severe joint pain with a longer pain duration than their male counterparts [10]. Similarly, data from a 2014 outbreak in Colombia showed that nearly half of the monitored symptoms were more frequent in female patients than in male patients [11]. In addition, chronic edema and arthralgia were significantly higher in female patients during an outbreak in the Dominican Republic [12]. These studies, as well as those from the recent SARS-CoV-2 pandemic, highlight the importance of considering sex as a biological variable when assessing viral immune responses; however, no studies have directly addressed the mechanism behind the obvious sex bias exhibited during CHIKV infection. In this review, we discuss the current understanding of the field as well as where future research efforts should focus more time and resources.

## 2. Innate Immune Response to CHIKV

CHIKV infection begins with a bite from an infected mosquito. Mosquito saliva acts as a transport medium for CHIKV to enter the body and begin replication at the inoculation site. Once the virus enters the skin, CHIKV infects and replicates in local cells such as fibroblasts, macrophages, and epithelial cells [13,14]. CHIKV successfully enters these cells via receptor-mediated endocytosis (Figure 1); however, other mechanisms of viral entry, such as macropinocytosis and cell-to-cell transfer of exosomes, are implicated in alphavirus entry into host cells (Figure 2) [2,13,15,16]. CHIKV then invades the subcutaneous capillaries and begins replicating in endothelial cells, fibroblasts, and macrophages [13]. From there, virions are transported to the secondary lymphoid organs by lymphocytes, where they continue to replicate in migratory cells and spread to the larger circulatory system. Via the circulatory system, CHIKV can infect target cells of the liver, muscles, joints, and, in severe cases, the brain.

### 2.1. Role Type I Interferons during CHIKV Infection

Interestingly, type I interferons (IFN-I) have been implicated as key components of viral clearance during CHIKV infection. IFN-I’s are a family of cytokines best known for their antiviral capabilities (named for their ability to “interfere” with viral replication). This cytokine family consists of IFN-α and IFN-β and is an integral part of the innate immune response. Almost all cells in the body have the capacity to produce IFN-I, typically after stimulation of the pattern recognition receptors (PRRs) of the innate immune system (discussed further in the next section). Downstream signaling from these PRRs, including the recruitment of transcription factors called interferon regulatory factors (IRFs), results in the production and secretion of IFN-I, which binds to the IFN-I receptor (IFNAR). The binding of IFN-I to IFNAR triggers numerous signaling pathways that induce interferon-stimulating gene (ISG) expression, enhance the activity of monocytes and dendritic cells, and promote B and T cell responses [17].

Following inoculation, CHIKV triggers a robust IFN-I response at the site of infection, limiting CHIKV infection in mammals, including humans [14,18,19]. One study found that monocytes exposed to CHIKV produced IFN-α in sufficient amounts to clear and control CHIKV replication [20]. In another study, mice lacking IFN-I receptors (IFNAR−/−) were unable to control CHIKV dissemination and ultimately succumbed to infection by day 3 post-challenge; however, this phenotype was reversed with bone marrow chimera from wild-type (WT) to IFNAR−/− [14]. Another group compared the response of mice deficient in IRF3, IRF7, or both IRF3 and IRF7 following infection with CHIKV. While the single knockout mice all survived with no significant differences from WT mice, the double knockout mice all succumbed to infection, though not as rapidly as IFNAR−/− mice [21]. Together, these studies highlight the significance of IFN-I signaling in controlling CHIKV infections in mammals.

### 2.2. Innate Immune Cells during CHIKV Infection

Monocytes and monocyte-derived macrophages are important members of the innate immune response due to their ability to phagocytose and destroy infectious agents; these cells also release pro-inflammatory cytokines that aid in the activation of other cell types. In the context of CHIKV, primary human and mouse macrophages are susceptible to CHIKV infection in vitro [16,22]. In vivo, macrophage-like synovial cells and fibroblast-like synoviocytes line synovial joints where patients generally experience pain [23,24]. A study using in vitro cultures of primary human synoviocytes demonstrated their susceptibility to CHIKV infection. Moreover, infection of human synoviocytes induced the secretion of pro-inflammatory cytokines such as IL-6, IL-8, MCP-1, and RANKL. Supernatants from CHIKV-infected synoviocytes also induced monocyte migration and differentiation of monocytes/macrophages into osteoclast-like cells that produce high levels of IL-6 and TNF-α, known arthritis mediators [23]. Monocytes and macrophages are proposed as reservoirs for persistent CHIKV in the chronic phase. Consistent with this proposal, analysis of a biopsy sample from one patient with chronic CHIKV infection revealed CHIKV RNA and proteins in perivascular synovial macrophages for 18 months post-infection. Therefore, macrophages may contribute to chronic CHIKV symptomology [25,26,27,28].

Dendritic cells (DCs) are another monocyte-derived cell type whose primary function in antigen presentation links the innate and adaptive immune systems. Though DCs have not been extensively evaluated for their role in CHIKV immunopathology, the existing data is conflicting. While human DCs in culture appear to be resistant to CHIKV infection, DCs from cynomolgus macaques are susceptible to CHIKV infection [16,28]. In addition, mice deficient in dendritic cell immunoreceptors (DCIR) develop more severe diseases than WT mice, including more rapid and severe CHIKV-induced edema [29]. This suggests that the DCIR on DCs plays an important role in limiting CHIKV-associated inflammation. Additionally, CHIKV RNA is reportedly detectable in splenic B cells and follicular DCs during the chronic phase of CHIKV infection, suggesting other potential cellular reservoirs for CHIKV persistence [30]. Plasmacytoid DCs (pDCs) are not directly infected by CHIKV; however, they still play an essential role in clearing CHIKV within mammalian hosts. One study developed a mouse model (based on IRF3/7−/−) in which IRF7 signaling was only present in pDCs. From this study, pDC-restricted IRF7 signaling was able to protect these mice from the lethal CHIKV challenge. In addition, the levels of IFN-I were undetectable [29]. Taken together, these studies suggest that DCs, while not directly infected by CHIKV, are important for its clearance.

### 2.3. Role of Pattern Recognition Receptors in Innate Immunity to CHIKV

Pattern recognition receptors (PRRs) are crucial for successful immune responses. PRRs are germline-encoded and recognize evolutionarily conserved pathogen-associated molecular patterns (PAMPs). Once activated, PRRs initiate signaling cascades that trigger the release of pro-inflammatory cytokines and chemokines, including IFN-I. PRRs are essential for host immunity because they release signals that attract immune cells to the site of infection; however, they also link the innate and adaptive immune responses by participating in antigen presentation to T and B cells for proper antibody production. Loss of PRRs can result in decreased neutralizing antibody responses and dysregulated recruitment of effector cells [31,32].

Three different PRR pathways are implicated in the induction of IFN-Is triggered by CHIKV. The first is a retinoic acid-inducible gene I (RIG-I) and melanoma differentiation-associated gene 5 (MDA5), which signal via mitochondrial antiviral signaling protein (MAVS, also called IPS-1 and Cardiff). RIG-I and MDA5 recognize single-stranded RNA (ssRNA) and double-stranded RNA (dsRNA) in the cytoplasm, making them ideal for recognizing intermediary dsRNA generated during CHIKV replication. Interestingly, one study found that RIG-I, but not MDA5, interacted with the CHIKV RNA genome at the 3′ untranslated region (3′UTR). This association was dependent on RNA secondary structures rather than nucleic acid content, which suggests that MDA5 may recognize different RNA PAMPs than RIG-I [33]. Interestingly, Cardif-deficient mice (or mice that lack downstream signaling of RIG-I and MDA5) become viremic but only have slight phenotypic differences from WT mice [14]. Because IFNAR−/− mice will die from CHIKV infection, these studies suggest that other signaling mechanisms must be involved in CHIKV-induced IFN-I production.

Toll-like receptor 3 (TLR3), another IFN-I signaling pathway, is implicated in CHIKV pathogenesis as well. TLR3 is the sole member of the TLR3 superfamily, and after recognition of dsRNA, TLR3 initiates signaling via toll-interleukin 1 domain-containing adaptor-inducing interferon- β (TRIF). Aside from IFN-I production, TLR3 is also required for proper antibody neutralization. Regarding CHIIKV infection, TLR3 agonist treatment inhibits CHIKV replication in vitro by upregulating IFN- β and other pro-inflammatory cytokines; this response also resulted in positive feedback on TLR3 expression [31]. Additionally, loss of TLR3 in CHIKV infection lowers the neutralization capacity of antibodies against CHIKV [32]. Taken together, TLR3 plays an essential role in proper innate and adaptive immunity to CHIKV.

Finally, TLR7 and TLR8 are possible pathways for IFN-I production during CHIKV infection. The TLR7 superfamily consists of TLR7, TLR8, and TLR9, and this superfamily utilizes myeloid differentiation primary response gene 88 (MyD88) as its signaling protein. TLR9 recognizes CpG DNA and is not implicated in CHIKV pathogenesis. TLR7/8 both recognize ssRNA as their ligand. Previously, TLR7/8 was thought to respond in the same way, but new evidence suggests that they each have distinct roles and cell tropisms. TLR7 is preferentially expressed in pDCs and B cells, whereas TLR8 is expressed in monocytes/macrophages, myeloid DCs (mDCs), and neutrophils [34]. Additionally, TLR7 and TLR8 are thought to maintain balance in cells where they are co-expressed. For example, one study evaluated the differences between TLR7/8 in monocytes/macrophages and found that upregulation of TLR7 following mRNA silencing of TLR8 led to TLR7-induced NF-κB activation but not TLR8-induced NF-κB activation in monocytes. In addition, macrophages treated with TLR7 and TLR8 agonists resulted in NF-κB activation without an ISG response [34]. Together, these studies suggest distinct roles and localization of TLR7 and TLR8.

To date, TLR7/8 has not been directly evaluated in reference to CHIKV replication; however, studies have found indirect evidence of its involvement. In one study, MyD88-dependent TLR7 signaling protected mice from severe Ross River Virus (RRV)-induced disease. RRV is another arthritogenic alphavirus, and this study eludes that the mechanism of protection by TLR7 is conserved across the alphavirus genus [35]. Another group found that neutrophil extracellular traps (NETs) effectively controlled CHIKV infection via a mechanism dependent on TLR7 activation and reactive oxygen species (ROS) generation [36]. In agreement with this, oxidative stress is caused by TLR8-induced neutrophilic responses but not TLR7-induced responses [36]. It’s important to note that studying TLR7/8 in neutrophils will differ between mice and humans. Whereas murine neutrophils express TLR7 predominantly, human neutrophils appear to only express TLR8. This provides strong evidence for studying these two PRRs as separate entities. Finally, a study using germ-free mice infected with CHIKV revealed that the addition of a single Clostridium symbiont restored the MyD88-TLR7 pathway, which in turn restored murine anti-CHIKV immunity [37]. Taken together, TLR7/8 appears to play a key role in the immune response to CHIKV, as well as other alphaviruses. Additionally, because TLR7/8 are both expressed on the X chromosome, they could play a role in the sex disparities exhibited during human CHIKV epidemics.

## 3. Adaptive Immune Response to CHIKV

While the innate response takes only hours to develop and begin, the adaptive immune response takes approximately one week to become activated. The adaptive immune system is generally thought to be made of two parts: humoral immunity from B cell antibody production and cell-mediated immunity from T cells and cytokine release. Previously, the two main branches of the immune system were thought to be separate entities; however, there are elements that bridge the two, suggesting a more complex system than previously thought. Antigen-presenting cells (APCs) make up an essential link between innate and adaptive immunity. Not only do APCs secrete cytokines (e.g., TNF-α, IL-6, IL-12, IL-18, IFN-γ, IFN-α/β), but they also contribute to clonal immunity by presenting processed antigenic peptides to T cells and B cells in draining lymph nodes. The most capable APCs are professional IFN-I-producing DCs because they can induce primary and secondary immune responses. DCs and cytokines are essential for acute clearance of infection, but they’re also valuable to B and T cell immune memory as well [38].

During human CHIKV infections, lymphopenia (or low levels of circulating lymphocytes) can occur acutely. CD8+ T cells will appear earlier in the immune response as cytolytic-inducing cells that release pro-inflammatory cytokines, paving the way for necessary adaptive responses. On the other hand, CD4+ T cells appear later and contribute directly to adaptive immunity. Some studies demonstrate that CD4+ T cells activate during chronic CHIKV disease to induce inflammation; this indicates immune-mediated damage as a possible mechanism for CHIKV symptomology [39]. Interestingly, CHIKV-infected RAG1-deficient mice (or mice that lack B and T cells) were able to clear infection acutely, though they took longer to recover than WT mice [40]. Additionally, T cell receptor (TCR)-deficient mice infected with CHIKV do not develop joint swelling, implying that T cell signaling may contribute to chronic CHIKV symptoms. Intriguingly, the joint pathology in these mice was recovered when mice were injected with CD4+ T cells from CHIKV-infected WT mice [41]. This study implies that adaptive immunity is less critical for controlling CHIKV infection than innate immunity, given that IFNAR−/− mice will consistently die from CHIKV infection.

## 4. Sex and Age as Biological Variables in CHIKV Immune Response

### 4.1. Sex as a Biological Variable

The mechanisms associated with sex-linked differences in immune response are still poorly understood; however, it is increasingly important to evaluate sex as a biological variable that affects immunity. For example, over 75% of autoimmune disorders occur in females; on the other hand, males are almost twice as likely as females to die from malignant, nonreproductive cancers [42]. Even during the SARS-CoV-2 pandemic, males were more likely to experience severe SARS-CoV-2 infection symptoms than female patients [43,44,45]. We know that females generally mount a more robust immune response than males, characterized by stronger IFN-I signaling and elevated antibody responses. Both sex hormones and sex chromosomal genetics have been implicated, but the exact mechanism remains unclear and has not been studied in the context of CHIKV.

Sex hormones contribute to how a person will respond to infections. Androgens (i.e., testosterone and dihydrotestosterone) reduce pro-inflammatory responses, as well as immune cell activity in general [42]. For example, males tend to produce less TNF, iNOS, and NO from macrophages, as well as express fewer surface TLR4s [46,47]. Additionally, IL-10 and TGF-β production is higher in males, which produces stronger anti-inflammatory responses [47,48]. This is confirmed in androgen-deficient males who produce more pro-inflammatory cytokines, higher antibody titers, and possess larger CD4/CD8 T cell ratios [42,49,50]. Castrated male mice will also produce higher CD4/CD8 T cell ratios and macrophage numbers [51]. Estrogen (i.e., 17β-estradiol), on the other hand, can increase the number of neutrophils in circulation and enhance the expression of PRRs on the surface of immune cells [42,52]. Estrogen has bipotential in monocytes and macrophages, where low concentrations increase pro-inflammatory cytokine production and high concentrations decrease them [53]. Estrogen also promotes the differentiation of monocytes and bone marrow precursor cells to inflammatory DCs and CD11c+ DCs, respectively [54,55]. These DCs, then, produce higher levels of IFN-α and express more TLR7; this phenotype is associated with greater internalization of pathogens and more antigen presentation [42,55]. Together, it appears that sex hormones could contribute to differences in immunity between males and females.

While sex hormones could be important, genetic mediators of sex are also possible contributors to sex differences in immune responses. The X chromosome houses many genes that are essential for immune system regulation. Females with two X chromosomes accomplish gene dosage compensation by epigenetically silencing one of the X chromosomes, seemingly at random, in every single cell [56,57]. This random inactivation is achieved by coating one of the X chromosomes in a long-noncoding RNA (lncRNA) XIST and produces a mosaicism of X chromosomes in females that could provide a genetic immune advantage over males [56]. Additionally, some genes of the X chromosome appear to be capable of escaping the transcriptional silencing, such as TLR7, TLR8, IRAK1, CYBB, BTK, and IL13RA1, leading to higher expression levels of these genes in females over males (Figure 3). In the case of TLR7, more gene expression of the PRR would allow for stronger signal transductions and more cytokine release. From these studies, genetic factors and hormonal attributes likely work together to create an evolutionary advantage for females in responding to infections.

### 4.2. Age as a Biological Variable

Age is another factor that contributes to how a person will respond to viral insults. CHIKV, specifically, is known to cause more severe disease in neonates and elderly patients; however, the immunopathogenesis of CHIKV has not been directly evaluated as it pertains to age [2,58,59,60]. In general, the immune response begins diminishing with age and is generally associated with decreased levels of sex hormones. The exact mechanism behind age-related immunity loss is also not completely understood. Some examples of this weakened immunity include a decrease in the number and function of circulating DCs in older adults [61]. In addition to fewer DCs, neutrophils exhibit impaired chemotactic ability, lower phagocytic capacity, and less NET formation as age increases [62,63]. NK cells from aged organisms also reveal decreases in cytotoxicity and IFN-γ and granzyme B production [62,64,65]. Macrophages also have lower phagocytic capacity and declined nitric oxide production [62,66]. PRR expression is also affected by age. TLR7 in pDCs and TLR8 in mDCs is decreased in aged animals [62,67,68]. When these studies are taken together, age and sex appear to contribute to disease outcomes after viral infection and should be considered biological factors when studying the pathogenesis of CHIKV.

## 5. Contribution of Mosquito Saliva to CHIKV Pathogenesis

As previously stated, CHIKV is a mosquito-borne pathogen primarily spread by *Aedes* species; however, much of what is known about the immunological response to CHIKV infection referenced above is from needle inoculation studies, which do not recapitulate normal human infection. In fact, few studies have evaluated the effects of mosquito salivary proteins (MSPs) on CHIKV pathogenesis. Mosquito saliva is known to induce a distinct host response because the mosquito has evolved to allow for quick feeding; this quick feeding requires specific factors to prevent the host immune response from blocking their access to a blood meal via vasoconstriction or platelet aggregation. Sialokinins, for example, are present in mosquito saliva and prevent vasoconstriction by increasing arteriole and vacuole size; sialokinins also appear to have an immunomodulatory effect on T cells. Aegyptin and apyrase are two MSPs that prevent platelet aggregation using different mechanisms; aegyptin binds host collagen, while apyrase inhibits ADP-dependent platelet aggregation [69,70]. There are also MSPs with direct effects on host immune responses (summarized in Table 1). One example of this is a 34kDa protein found in Ae. aegypti saliva [69]. This protein decreases IFN-I signaling and IFN-γ expression, and it is upregulated in the salivary glands of CHIKV-infected Ae. aegypti females [71]. These are just a few examples of MSPs that allow a separate immune response compared to needle inoculation of CHIKV.

Mosquito-borne CHIKV infection begins when female mosquitoes insert their proboscis into the dermis of a mammalian host, where they continuously release virus-infected saliva in search of a blood meal. Once the blood vessel is reached, the MSPs prevent vasoconstriction and platelet aggregation, which increases blood vessel permeability. The disrupted endothelial barrier allows for the recruitment of innate immune effector cells, such as monocytes and neutrophils, to the bite site. In the meantime, the virus replicates in skin resident fibroblasts, monocytes/monocyte-derived macrophages, and endothelial cells. The infiltration of innate effector cells allows for new cellular hosts for the virus to infect and spread via lymph circulation. Through the lymphatic vessels, the virus reaches draining lymph nodes and infects cells there. It is then able to enter the circulatory system to spread and infect other organs, such as the liver, spleen, muscles, joints, and sometimes the brain [69,70,72,73].

One of the first studies evaluating mosquito delivery versus needle delivery of CHIKV was performed in CD-1 mice. This group found a downregulation in Th1 cytokines but an upregulation in Th2 cytokines [74]. This alone suggests that the immune response to CHIKV is altered by mosquito delivery. Another study evaluated mosquito delivery of CHIKV in Swiss albino pups; they noticed higher morbidity and mortality, as well as higher skin viral RNA titers, viremia, peripheral organ invasion, and cellular infiltration in mice infected with CHIKV by mosquito bite [70]. An in vitro study using human skin fibroblasts found that when infected with CHIKV alone, IFN-I and ISGs were upregulated; however, when infected with CHIKV in the presence of *Ae. aegypti* saliva, these genes were downregulated [75]. Lastly, using humanized mice, one group inoculated CHIKV via mosquito bite delivery and found that these mice exhibited more human-like CHIKV disease; however, needle inoculation of the same mice did not reveal the same phenotype [76]. Additionally, the immune responses for each delivery type were distinct, further establishing that mosquito delivery, or at least infection with MSPs, should be the standard for studying CHIKV and other mosquito-borne arboviruses.

**Table 1 viruses-15-01869-t001:** Summary of references evaluating effects of mosquito saliva. Mosquito saliva contains properties that assist in the acquisition of a blood meal. Some of these protein components also interact with host immune responses, which mosquito-borne viruses have evolved to take advantage of. The table above summarizes some of the studies that evaluate the mammalian host immune response to viral insult in the presence of mosquito salivary components. Abbreviations: CHIKV—Chikungunya virus, SFV4—Semliki Forest Virus, DENV—Dengue virus, SubQ—subcutaneous.

Host Species	Virus (Strain If Provided)	Vector Species + Method	Impact on Host Response	References
Swiss albino pups	CHIKV (DRDE-06)	*Ae. aegypti* bite + SubQ needle inoc.	increased morbidity and mortality	[71]
			higher skin viral RNA load and viremia	
			more extensive viral dissemination	
			dysregulated cytokine release	
			enhanced cellular infiltration	
BALB/c mice	SFV4 (pCMV-SFV4)	*Ae. aegypti* bite + SubQ needle inoc.	extensive edema (viral inoculum retained at bite site)	[73]
			transient increase in skin neutrophils	
CD-1 mice (2 weeks)	CHIKV (LR 5’GFP)	*Ae aegypti* bite inoculation	suppression of Th1 cytokines, enhancement of Th2 cytokines	[74]
			downregulation of TLR3 expression	
			downregulation of IFN-gamma expression	
Human dermal fibroblasts	CHIKV (LR2006_OPY1)	*Ae. aegypti* saliva	decreased ISG expression	[75]
			downregulation of STAT2 and pSTAT2	
			increased CHIKV titer over time	
Hu-NSG Mice	CHIKV (37997)	*Ae. aegypti* bite inoculation	exhibit more human-like CHIKV symptomology	[76]
			increased circulating CHIKV RNA	
			increased dissemination over needle-inoculation	
			differential cell recruitment compared to needle inoculation	
Human Keratinocytes	DENV	recombinant *Aedes* salivary proteins (putative 34 kDa)	(1) decreased IFN-I mRNA expression	[71,77]
			(2) decreased IRF3 and IRF7 mRNA expression	
			(3) suppressed antimicrobial peptide expression	
			(4) increased viral RNA titers	

## 6. Gaps in Knowledge

Even with the progress made over the past 70 years, there is much to learn about CHIKV disease in humans. As climate change is actively expanding, more regions will be affected by *Aedes* species mosquitoes, and as a result, *Aedes*-borne viruses, such as CHIKV, will pose an increasing risk to millions of humans worldwide. Little is known about the host response to vector-transmitted CHIKV because much of the literature disregards the contributions of vector saliva to viral pathogenesis. Ignoring the mosquito component of this disease presents misleading data on the immunological response of the host to viral insults. Without a proper understanding of the effects of natural transmission on symptomology, prevention and treatment for CHIKV are delayed. To combat this, mosquito bite delivery (or co-inoculation of the virus with mosquito saliva) should be incorporated as a standard practice when studying pathogens such as CHIKV. It is important to recognize this as a challenge. Mosquito saliva is difficult to extract, and mosquito bite delivery blocks access to standardized titer inoculation. One option is to allow mosquitoes to feed in a specific region of the animal model, followed by needle inoculation with the virus in the same feeding region. While this model does not exactly mirror the natural route of infection, it allows natural feeding as well as consistent viral titer inoculation.

In addition to the vector component, sex, and age should be established as biological variables when studying the immune response. With age, sex hormone levels change in males and females throughout their lives. Some immune genes have sex hormone response elements, suggesting that their immune responses are affected by age. Moreover, the X chromosome encodes several immune genes that are critical for the immune response. Although one X chromosome in females should be transcriptionally silenced, some genes, including TLR7, TLR8, and IRAK1, escape this mechanism and become overexpressed. Consequently, these genes can be upregulated in female patients, making them better equipped to respond to immune insults. This escape from transcriptional silencing contributes to the sex bias displayed in autoimmune diseases but has not been extensively evaluated regarding viral infection and especially the delivery of viral infections via mosquito bite. Numerous studies have established a sex bias associated with chronic CHIKV symptoms in humans; however, there is no literature that explains this sex disparity mechanistically. Additionally, neonates and elderly CHIKV-infected patients are more likely to report severe disease manifestations owing to their immunocompromised status. Establishing how sex and age contribute to disease outcomes is crucial, especially in the era of precision medicine.

## Figures and Tables

**Figure 1 viruses-15-01869-f001:**
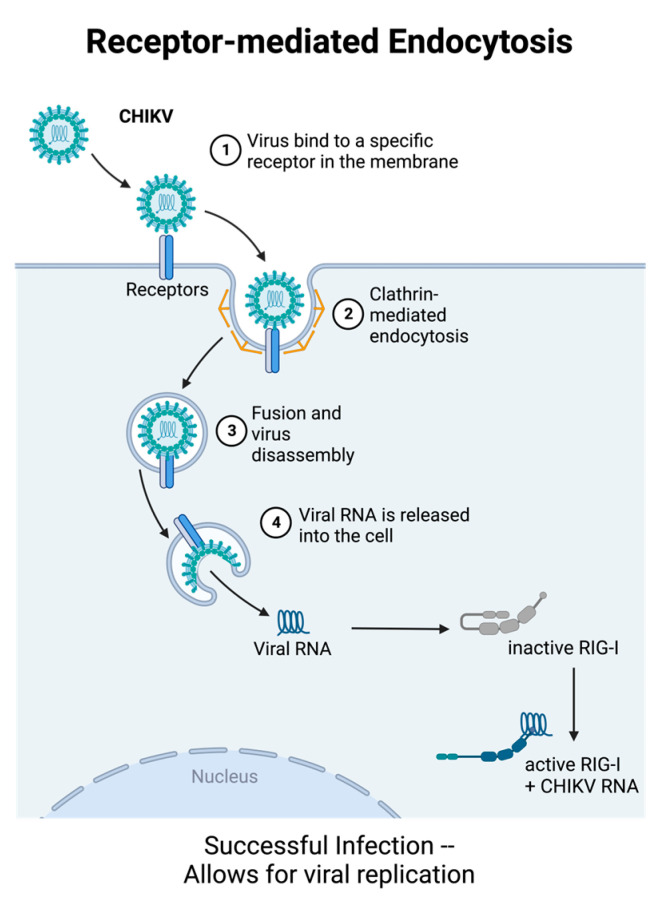
CHIKV begins successful replication by entering host cells via receptor-mediated endocytosis. Once packaged inside the host endosome, the pH drops, causing viral proteins to change conformation and fuse with the endosomal membrane. Once the viral proteins have fused with the endosome, the nucleocapsid is released into the cytoplasm and quickly degraded, releasing the viral genome into the host cytosol. In the cytosol, viral RNA can be recognized by RIG-I-like receptors (RLRs), such as RIG-I and MDA5. These pathways then trigger the production and secretion of type I IFNs. Created with Biorender.com (accessed on 31 July 2023).

**Figure 2 viruses-15-01869-f002:**
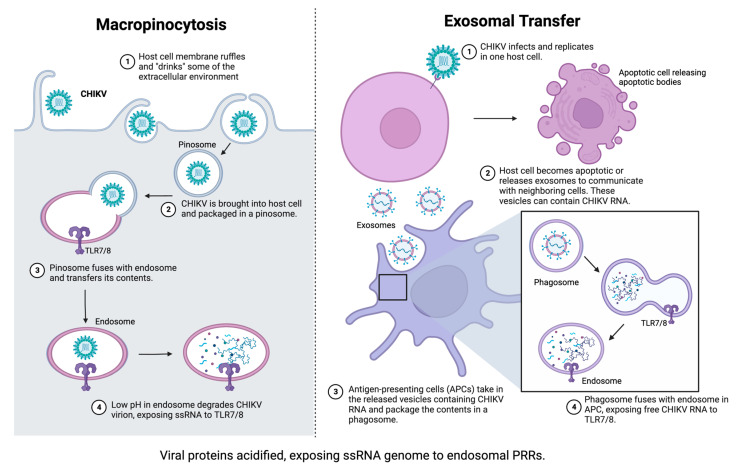
While receptor-mediated endocytosis seems to be required for successful CHIKV replication, other pathways of entry have been implicated for CHIKV and other alphaviruses. These alternative entry methods may not allow CHIKV to further replicate; however, they are still crucial for host immunity against CHIKV. These methods expose endosomal PRRs to CHIKV RNA, triggering different immune pathways than CHIKV-infected cells are capable of. In CHIKV-permissive cells, RIG-I/MDA5 would trigger IFN-Is, whereas, in the alternative methods, TLR7/8 would become activated instead. Created with Biorender.com (accessed on 31 July 2023).

**Figure 3 viruses-15-01869-f003:**
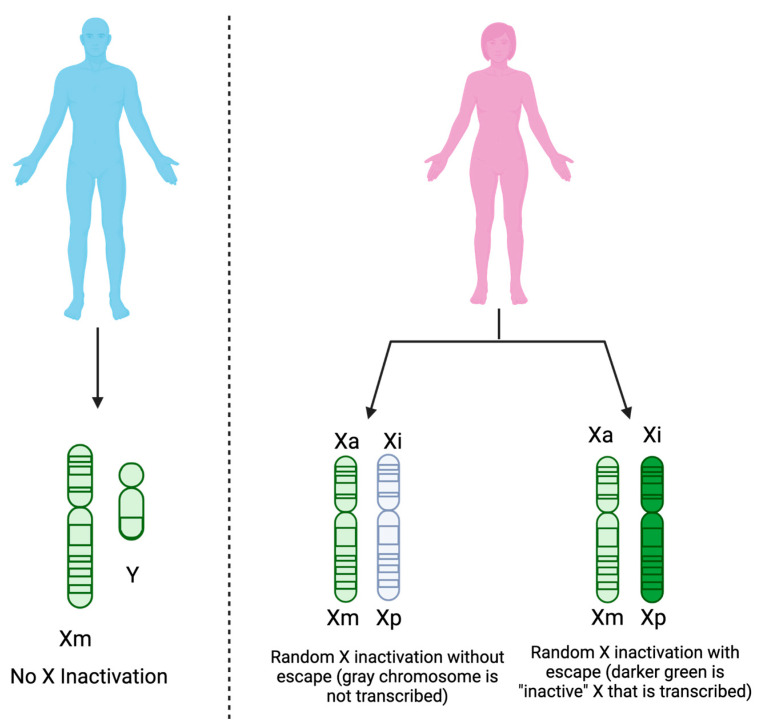
Gene dosage compensation is maintained across sexes via XCI in females. Because males have one X chromosome from the mother (Xm) and one Y chromosome from the father, XCI does not occur. Females, on the other hand, have two X chromosomes, one from the mother (Xm) and one from the father (Xp). One of the X chromosomes in females remains active (Xa), while the other is randomly inactivated (Xi) and generally remains that way (shown in grey). Recent studies, however, demonstrate the capability of some genes of Xi to escape this random inactivation, leading to increased gene expression in females (shown in dark green). Created with biorender.com (accessed on 31 July 2023).

## Data Availability

Not Applicable.

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
