# Peer review of "Immune Response to Chikungunya Virus: Sex as a Biological Variable and Implications for Natural Delivery via the Mosquito"

_viruses, 2023, doi:10.3390/v15091869_

Round 1

Reviewer 1 Report

Very rarely do I not have at least several suggestions or criticism when reviewing manuscripts, but this is an exception.  The manuscript is VERY well written, comprehensive relative to being a minireview, and, I think, presents quite a valuable overview of this area.  I also appreciated the sections on X-chromosome inactivation and how it might relate to differences among sexes in susceptibility to infectious diseases - a majority of virologists re likely not familiar with this phenomenon.

My only comment is that I wonder if the title is too restrictive.  Yes, you did have a section on sex as a biological variable for CHIKV infection, but the discussion of the effect of mosquito saliva on immune responses also stood out.  Perhaps "Immune Response to Chikungunya Virus: Sex as a Biological 2 Variable and the Effect of Mosquito Saliva" (yes, that's terrible, but at least think about expanding the title).

Author Response

Comment 1: “My only comment is that I wonder if the title is too restrictive.”

Response: Thank you for pointing this out. We completely agree, especially given the importance of mosquito saliva to pathogenesis. Therefore, we have changed the title to “Immune Response to Chikungunya Virus: Sex as a Biological Variable and Implications for Natural Delivery via the Mosquito” to reflect the premise of the paper more accurately.

Please see attached for full response to reviewers

Reviewer 2 Report

To the Authors:

The review entitled: “Immune Response to Chikungunya Virus: Sex as a Biological Variable” by Taylor and Rayner is interesting and draws attention to a neglected area of study (biological sex and/or age vs CHIKV infection).

Minor comments:

The authors should consider rephrasing some of the statements throughout the manuscript to reduce potential confusion by the reader. For example, line 12: “While female sex appears to be a risk factor…”, this could be interpreted as female sexual intercourse. It might be more appropriate to say “While the female sex appears to be a risk factor…” or “Females may be at an  increased risk…” or something to that effect. This goes for male and female “sex” throughout the manuscript. 

Line 11: “…show the importance…” show the importance or draw attention or demonstrate the importance?

Line 22-23. “…hosts through the bite of an infected mosquito…” Based on the mosquito species listed, this is the urban cycle. Different mosquitoes would be involved in a sylvatic cycle. 

Line 27: “As climate change continues to worsen…” This is a blanket statement. How will climate change impact CHIKV specifically? This sentence also needs a reference. 

Line 44-48. These sentences could be rewritten to improve clarity. 

Line 315 “Contribution of Mosquito Saliva to CHIKV Pathogenesis”. This section doesn’t really tie into the focus of the review, which is patient biological sex and/or age vs CHIKV infection. Recommend removing this section or adding to this section to tie it into the main premise of the review. 

Author Response

Comment 1: “For example, line 12: “While female sex appears to be a risk factor…”, this could be interpreted as female sexual intercourse.”

Response: You have raised an important point here; however, we believe that the use of “sex” is appropriate for the scope of this paper. The NIH describes “sex” as one’s biological factors, whereas “gender” refers to one’s psychological identity.  As such, the use of “sex” throughout should not be confused with sexual intercourse because intercourse is not mentioned throughout the manuscript. I’ve included a link to the NIH’s Office of Women’s Health for your reference.

https://orwh.od.nih.gov/sex-gender

Comment 2: “Line 11: “…show the importance…” show the importance or draw attention or demonstrate the importance?”

Response: We agree. We have, accordingly, revised the phrasing of this sentence to emphasize this point. The sentence now reads “In this review, we draw attention to the importance of studying sex as a biological variable by introducing epidemiological studies from previous CHIKV outbreaks.” This change can be found on lines 11-12 on page 1.

Comment 3: “Line 22-23. “…hosts through the bite of an infected mosquito…” Based on the mosquito species listed, this is the urban cycle. Different mosquitoes would be involved in a sylvatic cycle.”

Response: Thank you for pointing this out. We have changed the sentence to read “Chikungunya virus (CHIKV) is a mosquito-borne arbovirus that spreads to mammalian hosts through the bite of an infected mosquito, predominantly Aedes aegypti and Aedes albopictus in urban cycles [1–3].” These changes can be found on lines 23-25 on page 1.

Comment 4: “Line 27: “As climate change continues to worsen…” This is a blanket statement. How will climate change impact CHIKV specifically? This sentence also needs a reference.”

Response: Thank you for pointing this out. We have changed to sentence to read “As climate change continues to increase temperatures globally, the regions in which Aedes aegypti and Aedes albopictus can thrive are expanding [4].” We’ve also added a reference to this sentence, as well. These changes can be found on lines 28-30 on page 1.

Comment 5: “Line 44-48. These sentences could be rewritten to improve clarity.”

Response: Based on the reviewer’s comments, we have changed the sentence to improve clarity. It now reads “Though the first documented deaths of CHIKV appear to suggest worsening symptomology associated with its spread to previously unexposed areas, reporting biases also play a role. For example, CHIKV outbreaks tend to receive more attention than endemic cases of CHIKV. As a result, the areas from which CHIKV originated, such as the African continent, have not been as thoroughly monitored. This presents a large gap in knowledge regarding vector-borne disease transmission and symptomology and should be addressed expeditiously.”

Comment 6: “Line 315 “Contribution of Mosquito Saliva to CHIKV Pathogenesis”. This section doesn’t really tie into the focus of the review, which is patient biological sex and/or age vs CHIKV infection. Recommend removing this section or adding to this section to tie it into the main premise of the review.”

Response: Thank you for this suggestion. You’ve raised an important point here; however, we believe that the inclusion of the “Contributions of Mosquito Saliva” section is crucial for understanding the biases in data associated with CHIKV infection and was vigorously supported by two other reviewers. We do, however, agree that the title is restrictive and has been changed to reflect the true premise we hope to convey.

Please see attached for full list of response to reviewer comments

Reviewer 3 Report

The manuscript is an intensive review of immunological and demographic variables to be considered during a CHIKV infection. The text is impeccable and well-written, but since the article title is fundamentally about the variable sex, I believe that the authors should show more of those differences with some studies published. 

Author Response

Comment 1: “The text is impeccable and well-written, but since the article title is fundamentally about the variable sex, I believe that the authors should show more of those differences with some studies published.”

Response: Thank you for pointing this out. It would’ve been interesting to explore this aspect. However, in the case of CHIKV, we aren’t aware of any studies that directly evaluate the effect of biological sex. As a result, we tried to incorporate other studies from adjacent fields and apply it to CHIKV. We understand that the title was too restrictive, though, and have changed it to more accurately reflect the message we hope to convey.

Please see attached for full list of response to reviewer comments
